# GenNBV: Generalizable Next-Best-View Policy for Active 3D Reconstruction

## Abstract

Even with the recent advances in neural radiance rendering (NeRF) enable high-quality digitization of large-scale scenes, the image-capturing process is time-consuming and labor-intensive. Previous works attempt to automate this process using active 3D reconstruction, with the Next-Best-View (NBV) policy being the most well-known. However, the majority of NBV policies are rule-based and only apply to a predefined limited action space, limiting their generalization ability. In this work, we propose *GenNBV*, a novel framework that endows the first free-space NBV policy with generalizability through end-to-end training. This policy is reinforcement learning (RL)-based and empowers a 3D scanning drone to capture from any viewpoint and interact with the environment across diverse scenarios, even those involving unseen structures during training. We also proposed a novel scene representation using action, geometric, and semantic embeddings, to further boost generalizability. To evaluate this NBV policy, we also establish a benchmark using the Isaac Gym simulator with the Houses3K and OmniObject3D datasets. Experiments demonstrate that our approach achieves a 98.26% and 83.61% coverage ratio on unseen buildings from these datasets, respectively, outperforming prior solutions.

## 1 Introduction

Recent advances in 3D reconstruction (Park et al., 2019; Sun et al., 2021; 2022b) and neural rendering (Mildenhall et al., 2020; Sun et al., 2022a; Zhang et al., 2022) have significantly enhanced the quality of 3D digitization of large-scale scenes, such as buildings and city landmarks (Hardouin et al., 2020; Zhang et al., 2021; Liu et al., 2022; Xiangli et al., 2022; Li et al., 2023). To reconstruct a large-scale scene, the conventional approaches need professional pilots to skillfully navigate drones to capture multi-view images that can cover the majority. However, it typically takes a professional team several days' effort, thus limiting its scalability. Moreover, even professional pilots may miss some parts of a building at the first scan, leading to extra time or multiple rounds for rescanning. Such repetitive work exacerbates the already time-consuming and labor-intensive process.

To alleviate the manual effort in 3D scanning, active 3D reconstruction algorithms have emerged as a promising approach to automate flight path design and viewpoint selection Devrim Kaba et al. (2017); Chen et al. (2019); Chaplot et al. (2020). These algorithms enable robots to actively explore and interact with unseen environments. They alternate between inferring optimal viewpoints, capturing new data, and updating the rebuilt 3D model. The pioneering work is the Next-Best-View (NBV) policy for reconstruction, which determines the sequence of optimal viewpoints for scanning (Lee et al., 2022; Pan et al., 2022; Zhan et al., 2022; Peralta et al., 2020b; Guedon et al., 2022; Ran et al., 2023). With the optimal key viewpoints selected by NBV policy, a continuous piloting path can be generated using the classic planning modules, such as RRT (LaValle & Kuffner Jr, 2001) and FMM (Sethian, 1996). However, existing NBV algorithms fail to generalize due to two key aspects: 1) *Action space.* Many previous methods select views from a limited action space, such as a set composed of only one hundred candidate viewpoints (Pan et al., 2022), or a tiny pre-defined space like a hemisphere (Lee et al., 2022; Zhan et al., 2022; Guedon et al., 2022; Ran et al., 2023). Nevertheless, these predefined view sets may not apply to complicated scenes and thus reduce generalizability. 2) *Policy.* Existing approaches like uncertainty-driven works (Lee et al., 2022; Zhan et al., 2022; Ran et al., 2023) typically yield the NBV from hand-crafted rules from the scene representation. These methods tend to overfit specific scenes and may fail to generalize.

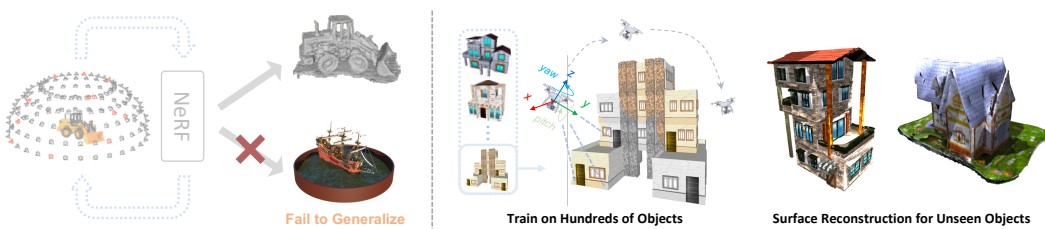

Figure 1: To determine the best view for 3D reconstruction, previous methods only chose from predefined views using manually defined rules on neural radiance field, lacking the ability to generalize to unforeseen scenes (Left). With our end-to-end trained, generalized free-space policy, it can generalize to unseen objects, enabling the captured drone to image from any viewpoint (Right).

In this work, we propose *GenNBV*, the *first* framework that endows the free-space NBV policy with generalizability through end-to-end training. First, for action space, we formulate a free space composed of both the drone's position and heading to make it capable of scanning a scene from any viewpoint. In this action space, the agent drone interacts with the training environment by sequentially collecting sensory information for a target building from viewpoints generated by an end-to-end policy. Second, we proposed an end-to-end trainable and generalizable policy. Specifically, after each capture, we calculate a coverage ratio Isler et al. (2016); Peralta et al. (2020b) from the reconstructed result and use it as a key component in the reward function. Once the coverage ratio reaches a threshold or a collision happens, the environment is reset and the building to be reconstructed is replaced. Leveraging the labeled trial-and-error experience, the policy is optimized using Reinforcement Learning (RL) to maximize the average reward across all structures in diverse shapes in the training stage. Consequently, the well-trained policy is capable of inferring near-optimal NBV when encountering unseen structures, enabling a zero-shot generalizable reconstruction.

Furthermore, we design a novel multi-modal generalizable scene representation to improve the robustness of the policy network. At each step, our policy network selects the NBV based on a latent state embedding derived from historical observations by the flying drone. Our multi-model state embedding includes an *action embedding* from viewpoint information, *semantic embedding* from the RGB images, and *geometric embedding* derived from a novel probabilistic voxel representation from multi-view depth maps. This hybrid scene representation conveys the uncertainty of remaining viewpoints and the completeness of the building reconstructed from the gathered viewpoints.

To validate the effectiveness of our method, we construct a dataset for training and evaluation with Houses3K (Peralta et al., 2020a) in the NVIDIA Isaac Gym (Makoviychuk et al., 2021) simulator and further test its generalization ability on novel building and object categories on OmniObject3D (Wu et al., 2023) and Objaverse (Deitke et al., 2023). In contrast to commonly used metrics in previous works, such as coverage ratio and number of views, we further propose to evaluate different methods with Area Under the Curve (AUC) of the coverage. It provides a comprehensive indicator to reflect the model's performance in both coverage sufficiency and view efficiency. Our method outperforms others by a large margin in terms of all metrics. We also provide visualization results and a demo video in the appendix to qualitatively evaluate our method.

## 2 RELATED WORK

**Traditional 3D Reconstruction.** Photometry and geometry are two crucial aspects of reconstruction evaluation. Neural implicit representation (Park et al., 2019; Mildenhall et al., 2020) has shown progress in photometric rendering performance but faces challenges like time-consuming optimization and poor generalization, hindering its application in real-time reconstruction problems such as 3D SLAM (Sucar et al., 2021; Sun et al., 2021; Ortiz et al., 2022; Zhu et al., 2022; Rosinol et al., 2022; Johari et al., 2023). Geometry, in contrast, typically corresponds to 3D representations such as point clouds and 3D mesh and is directly related to issues like collision avoidance. Considering these properties, researchers are prone to focus on geometric reconstruction in practical applications and we also follow this stream.

**Active 3D Reconstruction.** Active 3D reconstruction is a promising field that has not been thoroughly benchmarked yet. The pioneering work is the Next-Best-View (NBV) policy for reconstruction, which determines the sequence of optimal viewpoints for scanning (Lee et al., 2022; Pan et al., 2022; Zhan et al., 2022; Peralta et al., 2020b; Guedon et al., 2022; Ran et al., 2023). These works can be pivotally differentiated based on the paradigm of NBV policies: rule-based or learning-based. Existing rule-based approaches like uncertainty-driven works (Lee et al., 2022; Zhan et al., 2022; Ran et al., 2023) typically yield the NBV from hand-crafted rules from the scene representation, which tends to overfit specific scenes. Most learning-based policies (Chen et al., 2019; Chaplot et al., 2020; Peralta et al., 2020b) use a deep reinforcement learning algorithm like PPO (Schulman et al., 2017) to sequentially predict the optimal viewpoints based on observation. They must obtain feedback from task-related rewards such as coverage ratio (Peralta et al., 2020b; Guedon et al., 2022) and optimize during massive iteration with task environment.

The action space is designed based on the paradigm of NBV policies. Most rule-based NBV policies select views from a limited action space, such as a set composed of only one hundred candidate viewpoints (Pan et al., 2022), a tiny pre-defined space like a hemisphere (Lee et al., 2022; Zhan et al., 2022; Guedon et al., 2022; Ran et al., 2023), making it possible to overlook some important viewpoints due to unavailability. Even though learning-based category further explore the larger action space like a 2D plain (Chen et al., 2019; Chaplot et al., 2020) or a constrained 3D space (Peralta et al., 2020b), their limited action space still prevent them from capturing sufficient details for 3D reconstruction.

Scene representation built from history observations, which directly provide reconstruction progress to NBV policies, is also a crucial aspect of the active 3D reconstruction framework. Previous works have explored visual representations for NBV policies, such as TSDF (Hardouin et al., 2020), neural radiance field (Adamkiewicz et al., 2022), and 2D BEV maps (Chaplot et al., 2020; Ye et al., 2022; Guo et al., 2022). However, implicit representations are hard to jointly with learning-based NBV policies, and 2D BEV map lacks sufficient information for large-scale outdoor scenes that contain numerous geometric details in 3D space.

**Generalizable Reinforcement Learning.** Increasing the diversity of training data leads to better generalizability (Cobbe et al., 2020). In the robotics field, domain randomization (Tobin et al., 2017) allows legged robots to walk on various terrains absent from the training environment. For end-to-end driving policy, training in large-scale synthetic and realistic scenarios improves the safety of autonomous cars (Li et al., 2022b;a) in unseen test scenarios. Scan-RL (Peralta et al., 2020b) and SCONE (Guedon et al., 2022) are two pioneering works investigating the generalizability of active 3D reconstruction frameworks, while both of them suffer from the constrain of viewpoint sampling in limited space. In this work, we tackle the aforementioned issues by predicting NBV in free 3D space with a learning-based planner, which is trained with a dataset containing buildings in various shapes and poses for acquiring generalizability for unseen buildings.

## 3 METHODOLOGY

In this section, we deliver our active 3D reconstruction framework GenNBV, especially the pivotal Next-Best-View policy. An overview of our framework is illustrated in Fig. 2. Firstly, we formulate the NBV problem as a Markov Decision Process (MDP), with a novel design of observations (blue boxes in Fig. 2) and action space in Sec. 3.1. Next, we elaborate our end-to-end NBV policy $\pi$ (orange box in Fig. 2) in Sec. 3.2. Inspired by Curl (Laskin et al., 2020), we point out that generalizability greatly depends on how to extract embeddings (green box in Fig. 2), which reflect the reconstruction progress, from raw sensory observations like RGB images and camera poses. In Sec. 3.3, we introduce the reward function (right-hand side of Fig. 2) reflecting the optimization objective and the details of policy optimization.

### 3.1 FORMULATION OF THE NEXT-BEST-VIEW PROBLEM

We formulate the NBV problem as learning an optimal policy $\pi$ that controls the capturing process, such that enough information is captured for large-scale scene reconstruction, with limited decision-making budgets. As capturing, transferring, and processing a large set of captured images introduce significant computational costs, we also want to design a policy that also minimizes the number of

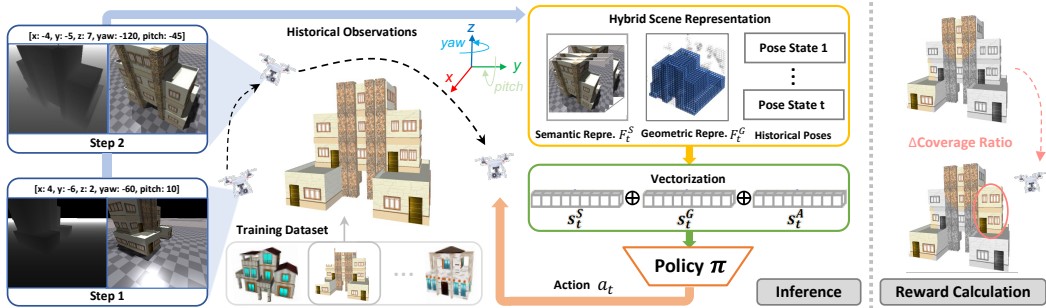

Figure 2: Overview of our proposed framework GenNBV. Our end-to-end policy takes the historical multi-modal observations as input, transforms them into a more informative scene representation, and produces the next viewpoint position. A reward signal will be returned at training time to optimize the end-to-end policy for maximizing the expected cumulative reward in one episode. Specifically, the signal is the increased coverage ratio after collecting a new viewpoint.

captured images. Therefore, our policy only captures sparse keyframes that sufficiently record all details of objects.

As shown in Fig. 2, at each time step $t$, the agent receives a visual observation $o_t$, takes an action $a_t$, infers the action at the next time step $t + 1$ to move to a new location, and then receives a new visual observation, repeating this interaction with the environment until the episode ends. Our simulated agent is embodied as CrazyFlie (Giernacki et al., 2017), a type of unmanned aerial vehicle equipped with various sensors, including an RGB-D camera and an IMU, to execute data collection for reconstruction. We discussed the details of the observation space and action space below.

**Observation Space.** As shown in the left column of Fig. 2, at each time step, the agent receives an RGB image $I_t$, a depth map $D_t$, and a state vector including the heading (yaw and pitch) and position (x, y, z) of the onboard camera. The observation $o_t$ (yellow box in Fig. 2) consists of all previously captured images in one episode, historical actions, and the current captures and actions. With this information as input, the policy network can estimate the progress of data collection and determine where to scan next.

**Action Space.** Unlike most previous NBV algorithms, we use larger action space in order to cover all details of objects. Specifically, we use the camera location and camera angle as our action space, which is a 5-dimension vector consisting of 3D position coordinates and 2D rotation angles (yaw-axis and pitch-axis). We restrict the roll-axis rotation as it is not supported in all drone platforms.

### 3.2 GENERALIZABLE STATE EMBEDDING

To ensure that the policy learned on one set of 3D objects can generalize to objects with different appearances and structures, a smart state embedding of raw sensor observations is needed to capture invariant features across different objects. Previous methods (Peralta et al., 2020b; Chaplot et al., 2020) only extract representations in 2D space, which are very sensible to appearance changes, and our experiments have shown that policies only trained on 2D features may not generalize to different objects. Instead, we propose a multi-model representation that has better generalizability.

Specifically, we first build two mid-level representations that can better model the relationship between the 3D object and our agent: a 3D geometric representation $F^G$ from depth maps and a semantic representation $F^S$ from RGB images. Then we encode these mid-level representations and concatenate them with a pose embedding into state embedding $s_t$, guiding the NBV policy for subsequent decision-making. The overall encoder for state embedding is shown in Fig. 2, and details of each representation are discussed below.

**Geometric Representation** One simple way to record the geometry of a 3D object is using a binary 3D occupancy map (Chaplot et al., 2020), where the value of each cell in the 3D cube indicates whether a cell contains a 3D object or not. However, since the 3D representation is gradually updated with newly captured data, this simple binary occupancy map cannot differentiate a real unoccupied

cell from an unscanned cell. An unscanned cell is a strong indicator that the agent should capture more data in this region, while no further scan is needed for a real unoccupied cell.

Previous works (Chaplot et al., 2020; Ye et al., 2022; Guo et al., 2022) simplify the 3D scene representation into a 2D Bird's Eye View map for actively reconstructing indoor scenes. However, the 2D BEV map lacks sufficient information for our large-scale outdoor scenes that contain many geometric details in 3D space. In addition, their wheeled robotic platforms are constrained to freely scan and represent these outdoor scenes. Therefore, to model the scanning process in 3D free space, we employ the probabilistic 3D grid (Thrun, 2002) as our geometric 3D representation, which records the probability of each 3D voxel being captured or not. Specifically, we first obtain a 3D point cloud in the world coordinate by back-projecting all 2D pixels to 3D points, using the depth map $D_t$, camera intrinsic parameters, and camera pose $a_t$. By voxelizing the obtained point cloud, we then build a 3D occupancy grid that explicitly indicates the binary state (occupied or free) in this 3D space. Subsequently, we represent this 3D grid as a probabilistic occupancy grid $F_t^G$ and extend the state space of voxels with three states (occupied, free, unknown).

During each scanning (one episode in RL), at each step $t + 1$, we update the probabilistic occupancy grid $F_{t+1}^G$ based on the preceding grid $F_t^G$ and current observation. Specifically, the grid is updated through Bresenham's line algorithm (Bresenham, 1965), which casts the ray path in 3D space between the camera viewpoint and the endpoints among the point cloud back-projected from depth $D_{t+1}$. Following the classical occupancy grid mapping algorithm (Thrun, 2002), we have the log-odds formulation of occupancy probability:

$$\log \text{Odd}(v_i|z_j) = \log \text{Odd}(v_i) + C, \tag{1}$$

where $v_i$ denotes the occupancy probability of $i^{th}$ voxel in the grid $F_t^G$, $z_j$ is the measurement event that $j^{th}$ camera ray passes through this voxel and C is an empirical constant. The derivation can be found in Appendix A.3. Thus, we update the log-odds occupancy probability of each voxel in the grid $F_t^G$ by adding a constant one time when a single camera ray passes through this voxel. Note that the probabilistic occupancy grid $F^G$ is continually updated within one episode. Finally, the occupancy status of voxels is classified into three categories: unknown, occupied, and free, using preset probability thresholds.

**Semantic Representation.** Geometric representation enables agents to comprehend spatial occupancy, yet it's insufficient for perceiving the environment. For example, when observing a hole in an object, the agent may struggle to differentiate between incomplete scanning and the actual presence of a hole in the object. In such cases, the semantic information contained in the captured RGB images can help the agent distinguish between these two scenarios.

To provide semantic information, we employ a preprocessing module that takes as input the current frame of RGB image $I_t$ and preceding $k$ frames and converts these frames $[I_t, I_{t-1}, ..., I_{t-k}]$ to grayscale, and concatenate them as output, following Peralta et al. (2020b). Then the preprocessed frames are fed into a two-layer convolutional network for extracting the semantic representation $F_t^S$.

**State Embedding.** To further combine the semantic and geometric embeddings, we first encode them to $s_t^S = f^S(F_t^S)$ and $s_t^G = f^G(F_t^G)$ where $f^*$ are learnable networks $\text{Linear}(\text{Flatten}(x))$, as shown in Fig. 2. Subsequently, we combine them with the historical action embedding $s_t^A = \text{Linear}(a_{1:t})$ to generate the final state embedding $s_t$, as the input to the policy network. This process can be formulated as: $s_t = \text{Linear}(s_t^G; s_t^S; s_t^A)$.

**Policy Network.** Taking the state embedding $s_t$ as input, the policy network is a 3-layer multilayer perceptron network (MLP) whose output is used to parameterize a normal distribution over action space. In this way, the action can be drawn from the stochastic policy $a \sim \pi(\cdot|o_t)$.

## 3.3 REWARD FUNCTION AND OPTIMIZATION

We train the end-to-end policy with reinforcement learning (RL) and hence design a reward function to precisely reflect the task objective for 3D reconstruction. The policy is optimized with proximal policy optimization Schulman et al. (2017) (PPO) for parallelizing sampling.

**Reward Function.** With the occupancy probability $F_t^G$ at time step $t$, we can threshold each voxel with an empirical bound to determine if it is occupied or not. This discrimination process outputs a

binary occupancy grid with $\tilde{N}_t$ voxels being occupied, which is used to calculate the coverage ratio:

$$\mathrm{CR}_t = \frac{\tilde{N}_t}{N^*} \cdot 100\%, \tag{2}$$

where $N^*$ is the number of ground-truth occupied voxels representing the surface of objects. To encourage our NBV policy to cover as many unseen views as possible, we use the difference of coverage ratio (CR) between two consecutive steps as the reward function $r^{CR}$:

$$r_{t+1}^{\mathrm{CR}} = \mathrm{CR}_{t+1} - \mathrm{CR}_t. \tag{3}$$

In free-space exploration, we also need to avoid collision. Previous limited-space agents (Lee et al., 2022; Zhan et al., 2022; Guedon et al., 2022) do not consider collision avoidance since their search space, like hemisphere, is by-design safe. Thus, we add a negative reward for collision and terminate the episode if a collision happens or the current coverage ratio is greater than 95%. We also implement a negative reward when the number of captured keyframes is over an empirical threshold to improve the path efficiency.

**Policy Optimization.** Once the reward function has been specified, the policy can be learned through any off-the-shelf RL algorithm. In this work, we specifically use PPO as thousands of workers can be parallelized to improve the sample efficiency. Specifically, given our parameterized policy $\pi_\theta$, the objective of PPO is to maximize the following objective function:

$$L(\theta) = \mathbb{E}_t \left[ \frac{\pi_\theta(a_t|s_t)}{\pi_{\theta_{\mathrm{old}}}(a_t|s_t)} A^{\pi_{\theta_{\mathrm{old}}}}(s_t, a_t) \right], \tag{4}$$

where $A^{\pi_{\theta_{\mathrm{old}}}}(s_t, a_t)$ is the advantage function that measures the value of taking action $a_t$ at state $s_t$ under the current policy $\pi_{\theta_{\mathrm{old}}}$. To prevent significant deviation of the new policy from the old policy, PPO incorporates a clipped surrogate objective function:

$$L^{\mathrm{CLIP}}(\theta) = \mathbb{E}_t \left[ \min \left( \eta_t(\theta) A^{\pi_{\theta_{\mathrm{old}}}}(s_t, a_t), \mathrm{clip}(r_t(\theta), 1 - \epsilon, 1 + \epsilon) A^{\pi_{\theta_{\mathrm{old}}}}(s_t, a_t) \right) \right], \tag{5}$$

where $\eta_t(\theta) = \frac{\pi_\theta(a_t|s_t)}{\pi_{\theta_{\mathrm{old}}}(a_t|s_t)}$ and $\epsilon$ is a hyper-parameter that controls the size of the trust region.

## 4 EXPERIMENTS

In this section, we conduct experiments on Houses3K Peralta et al. (2020a), OmniObject3D Wu et al. (2023), and Objaverse Deitke et al. (2023). For our policy and other learning-based policies, we train them on Houses3K training datasets. After this, we evaluate all policies on the Houses3K test dataset and the OmniObject3D dataset for quantifying the in-distribution and out-of-distribution generalizability. Finally, we show the visualization results on three datasets, demonstrating the effectiveness of the proposed method.

### 4.1 EXPERIMENTAL SETUP

**Simulation Environment.** We conduct all experiments in NVIDIA Isaac Gym (Makoviychuk et al., 2021), a physics simulation platform designed for reinforcement learning and robotics research. With GPU-accelerated tensor API and sensor interaction API, we can easily implement customized CUDA kernels to efficiently calculate our geometric representation. Moreover, the sensor simulation is efficient. It can run up to 1000 FPS, significantly reducing the training time. We create the agent drone CrazyFlie (Giernacki et al., 2017) and equip it with sensors including RGB-D cameras and IMU. Considering memory limitations, we downsample the image resolution to $400 \times 400$. The vertical field of view of this onboard camera is 90°. We also set up a point light source with a fixed position and constant intensity in Isaac Gym.

**Dataset.** We conduct our experiments on Houses3K (Peralta et al., 2020a), OmniObject3D (Wu et al., 2023) and Objaverse 1.0 (Deitke et al., 2023). Our model is trained on large-scale 3D objects from Houses3K. To validate the effectiveness and generalizability of our model, we evaluate it on *unseen* 3D objects from Houses3K. For testing the cross-dataset generalizability, we also evaluate it on batches of various objects from OmniObject3D and Objaverse datasets, which include both house and non-house categories.

Table 1: Evaluation results of Next-Best-View policies for active 3D reconstruction on **Houses3K** and the house category from **OmniObject3D**. The number of views is set to 30 and 20 for Houses3K and OmniObject3D, respectively. "*": the policy is trained with the Houses3K training set and evaluated with holdout Houses3K test set and OmniObject3D test set. "†": the ungeneralizable policy is directly trained and evaluated on corresponding test sets.

| Dataset | Houses3K | | OmniObject3D | |
|---|---|---|---|---|
| Policy | Mean AUC ↑ | Final Coverage Ratio ↑ | Mean AUC ↑ | Final Coverage Ratio ↑ |
| Random | 48.53% | 58.24% | 51.75% | 63.48% |
| Random Hemisphere | 71.19% | 79.72% | 63.70% | 72.91% |
| Uniform Hemisphere | 82.91% | 89.71% | 71.55% | 79.99% |
| †Uncertainty-Guided (Lee et al., 2022) | 83.13% | 89.31% | 69.48% | 79.93% |
| †ActiveRMap (Zhan et al., 2022) | 84.86% | 90.77% | 71.80% | 80.19% |
| *Scan-RL (Peralta et al., 2020b) | 87.39% | 95.63% | 74.75% | 80.63% |
| **\*GenNBV (Ours)** | **91.19%** | **98.26%** | **76.31%** | **83.61%** |

Houses3K contains 3,000 textured 3D building models. They are divided into 12 batches, with each batch featuring 50 distinct geometries and 5 varying textures. We further eliminate poorly designed geometries based on the following two criteria: 1) objects with complex internal structures (as we are mostly focusing on the surface reconstruction), and 2) objects with redundant bases at bottom. Following these criteria, the training set consists of 256 objects from 6 selected batches, and the test set consists of 50 objects from another batch. All these selected training and test objects have distinct geometries.

OmniObject3D offers a collection of 6K high-fidelity objects that are scanned from real-world sources across 190 typical categories. The house category from OmniObject3D for evaluation has 43 diverse objects. In contrast, Objaverse 1.0 comprises a vast library of 818K 3D synthetic objects spanning 21K categories. The diverse and high-quality nature of these datasets makes them ideal for evaluating our models and visualizing the results.

**Evaluation Metrics.** The objective of NBV policies is to capture the most useful information for reconstruction, with the least number of views. Most prior works (Peralta et al., 2020b; Guedon et al., 2022) use the coverage ratio (%) and the number of views to evaluate the performance of NBV planning policies. Coverage Ratio (%) (Isler et al., 2016) quantifies the geometric completeness of reconstruction, but is highly correlated with the number of views. Therefore, we propose to unify the number of views to a fixed value for all NBV policies during evaluation and use the area under the curve (AUC) of coverage ratio as the main metric for comparison. Thus we report the Final *Coverage Ratio* and *Mean AUC* along with the consistent number of views in all tables.

**Implementation Details.** We conduct all experiments in Isaac Gym simulation engine with one NVIDIA Tesla V100 GPU. Our implementation refers to the codebase of Legged Gym (Rudin et al., 2021) and the PPO implementation in Stable Baseline3 (Raffin et al., 2021). The ground-truth point clouds on objects' surfaces are sampled by the Poisson Disk sampling method (Yuksel, 2015) using Open3D API (Zhou et al., 2018). Please refer to the Appendix A for further details.

## 4.2 Performance Comparison

To comprehensively demonstrate the effectiveness and generalizability of GenNBV, we design three levels of evaluation experiments: 1) As shown in Table 1, we show the performance of our NBV policy on the Houses3K test set; 2) We evaluate the policy trained on Houses3K training set to the house category from OmniObject3D which has completely different geometric structures and textures compared to Houses3K training set. The quantitative result is at Table 1 and the visualization result is as shown in Fig. 3; 3) we also compare the generalizability of GenNBV with others by evaluations on an outdoor scene with enormous details from Objaverse to demonstrate the potential of GenNBV to generalize to the city-scale scene.

Table 2: Ablation studies of representations in our framework on Houses3K test set.

| | Representation Category | | | Evaluation Metrics | |
|---|---|---|---|---|---|
| | Probabilistic 3D Grid | 5-DoF Pose | RGB Image | Mean AUC ↑ | Final Coverage Ratio ↑ |
| Unimodal | ✓ | | | 81.06% | 84.56% |
| | | ✓ | | 69.53% | 76.61% |
| | | | ✓ | 81.24% | 87.90% |
| Multimodal | ✓ | ✓ | | 88.66% | 96.67% |
| | ✓ | | ✓ | 89.77% | 95.31% |
| | | ✓ | ✓ | 88.30% | 96.29% |
| | ✓ | ✓ | ✓ | **91.19%** | **98.26%** |

We implement the following six policies as our baseline algorithms. 1) **Random**: This policy randomly generates 5-dim vector $(x, y, z, pitch, yaw)$ among the action space as the next viewpoint. 2) **Random Hemisphere**: This policy randomly generates the next positions on a pre-defined hemisphere that sufficiently covers all objects of the test set. The headings are constrained to point to the center of the hemisphere. 3) **Uniform Hemisphere**: All positions are evenly distributed on the previously mentioned hemispheres. 4) **Uncertainty-Guided Policy** (Lee et al., 2022): This NBV selection policy iteratively selects the next view from a pre-defined viewpoint set according to the uncertainty based on a continuously optimized neural radiance field. 5) **ActiveRMap** (Zhan et al., 2022): This policy also adopts an iterative selection framework, with multiple objectives including information gain, to select the next viewpoints from the candidate set. 6) **Scan-RL**: This RL-based NBV policy predicts the next viewpoint only relying on the historical RGB images.

As shown in Table 1, GenNBV shows the best in-distribution and out-of-distribution generalizability in both coverage sufficiency and view efficiency, when evaluated on test sets consisting of unseen objects from Houses3K and OmniObject3D. Moreover, learning-based NBV policies such as GenNBV and Scan-RL, with much larger action space, outperform all rule-based baselines, even if ActiveRMap and Uncertainty-Guided baselines are trained and evaluated on the same 3D objects.

### 4.3 ABLATION STUDY

In Sec. 3.2, we introduce how to build the state embedding from diverse features. Here we reveal the importance of specific features with the ablation results shown in Table 2. Thanks to the same shape of features and state embedding, we only need to adjust the input dimension of the linear layer of state embedding according to the modal type when ablating the model.

To investigate the effect of our multimodal representation, we also evaluate the effect of different combinations of representation categories in Table 2. In unimodal experiments, we design the 5-DoF Pose baseline (i.e. Prior Trajectory) that learns an empirically optimal trajectory, which helps understand the effectiveness of prior knowledge in active 3D reconstruction. We demonstrate the effectiveness of our multimodal representation for policy learning in multimodal experiments.

### 4.4 QUALITATIVE RESULTS

We visualize the reconstruction results generated from the scanning trajectory of a single episode in Fig. 3. It demonstrates that our next-best-view policy can reconstruct objects better in terms of completeness and appearance quality compared to Scan-RL. We further visualize the reconstruction results of an outdoor scene from Objaverse using 30 collected views in Fig. 4.

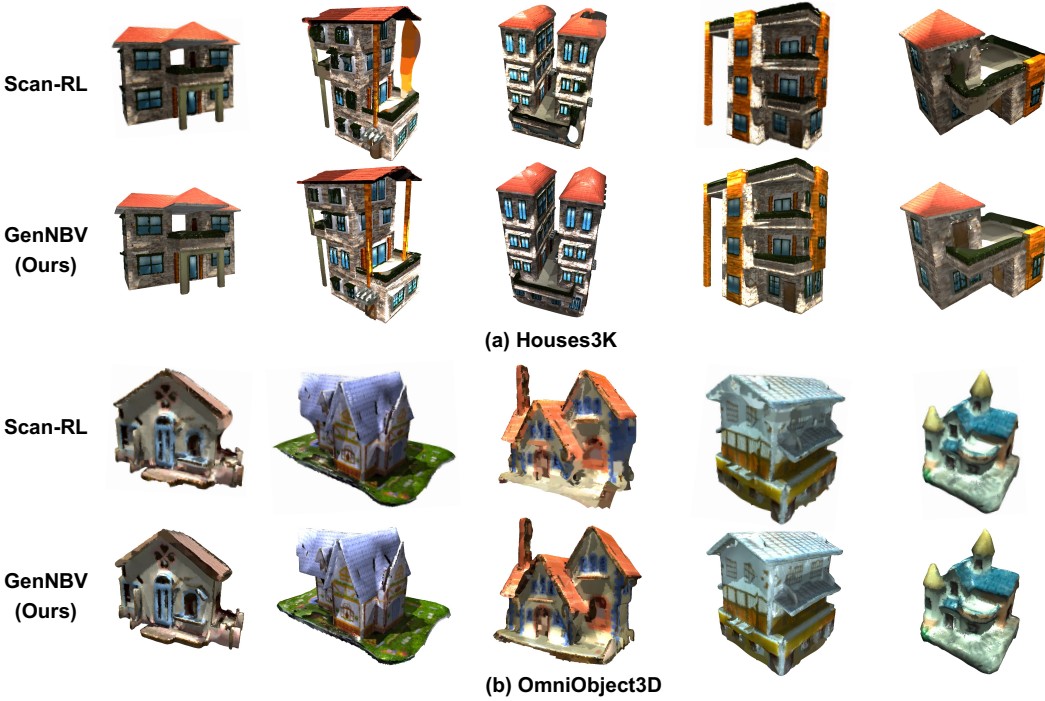

**(a) Houses3K**

**(b) OmniObject3D**

Figure 3: The visualization results of unseen 3D objects reconstructed by Scan-RL (Peralta et al., 2020b) and our model to compare the generalizability. (a) Unseen buildings from the test set of Houses3K. (b) Unseen buildings from OmniObject3D. It is obvious that some parts of the model reconstructed by Scan-RL are wrong or missing. For example, the second object in the first row has a pillar in the wrong shape. Scan-RL fails to reconstruct the roof edge for the fourth object from OmniObject3D, as shown in the third row.

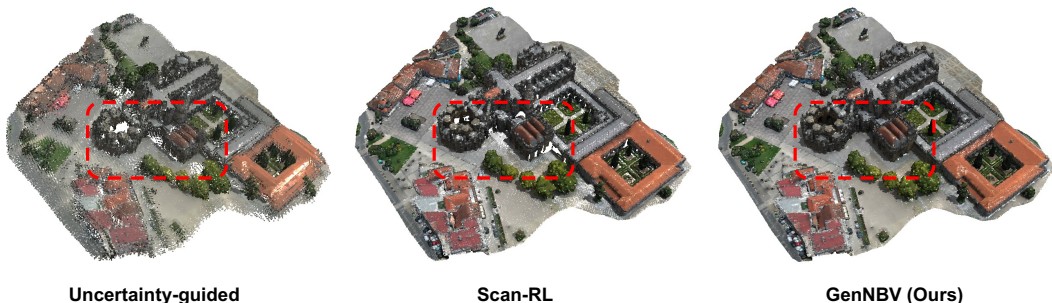

**Uncertainty-guided**          **Scan-RL**          **GenNBV (Ours)**

Figure 4: The visualization results of an unseen 3D outdoor scene with enormous details from Objaverse, reconstructed by Uncertainty-guided, Scan-RL and our model. Compared to the uncertainty-guided method and Scan-RL, the scene reconstructed by our method is more watertight and has fewer holes on the ground and building surface, especially in the region highlighted by the red box.

## 5 CONCLUSION

This study presents an end-to-end approach for active 3D scene reconstruction, reducing the need for manual intervention. Specifically, the learning-based policy explores how to reconstruct diverse objects in the training stage and thus can generalize to reconstruct unseen objects in a fully autonomous manner. Our controller maneuvers in free space and the next best view selection based on a hybrid scene representation which conveys scene coverage status and thus reconstruction progress. We show the effectiveness of our approach by testing it on multiple datasets including Houses3K, OmniObject3D, and Objaverse. The quantitative and qualitative results on Houses3K and Objaverse show that our method outperforms other baselines in terms of in-distribution and out-of-distribution generalizability. In addition, the experiment conducted on Objaverse shows that the policy trained in single-building settings can even generalize to a large-scale outdoor scene.

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
