## A APPENDIX

### A.1 DEMO VIDEO

To show how the trained end-to-end policy selects the most informative NBV and comparisons with other baselines, we provide a demo video at https://drive.google.com/file/d/1pvT0nOr8f7lD2bPQsR50DZTVDvtIz81U/view?usp=sharing

### A.2 IMPLEMENTATION DETAILS OF BASELINE POLICIES

The implementation details of baseline policies are described below:

1) **Random Policy**: This policy randomly generates 5-dim vector $(x, y, z, pitch, yaw)$ among the action space as the next action. The randomly generated positions are constrained so as not to cause collisions.

2) **Random Policy on the Sphere**: This policy randomly generates positions $(x, y, z)$ on a pre-defined hemisphere that exactly covers all objects of the test set. The headings are required to point to the center of the hemisphere.

3) **Uniform Policy on the Sphere**: All positions are evenly distributed on the previously mentioned hemispheres. Specifically, all sampling points are distributed over 5 heights, each with 10 evenly spaced positions.

4) **Uncertainty-Guided**: We use TensoRF (Chen et al., 2022) as the implementation foundation of neural radiance field. Before implementing uncertainty-driven viewpoint selection, we sample 100 views as the whole candidate set on a pre-defined hemisphere.

5) **ActiveRMap**: We implement the "discrete (free)" setup of ActiveRMap, which constrains the drone agent on the pre-defined hemisphere.

6) **Scan-RL**: In order to fairly compare the former Next-Best-View policy Scan-RL (Peralta et al., 2020b) with us, we implement Scan-RL in our experimental setup, with our action and state space. Also, we replace its optimization algorithm in Scan-RL with PPO, which experimental results show that can achieve better coverage.

### A.3 DETAILS OF OCCUPANCY GRID MAPPING ALGORITHM

Before updating the probabilistic occupancy grid $F_t^G$, Bresenham's line algorithm is implemented to cast the ray path in 3D space between the camera viewpoint and the endpoints among the point cloud back-projected from $D_{t+1}$. According to the classical occupancy grid mapping algorithm (Thrun, 2002), we have the log-odds formulation of occupancy probability:

$$\log Odd(v_i|z_j)) = \log Odd(v_i) + \log \frac{p(z_j|v_i = 1)}{p(z_j|v_i = 0)}, \tag{6}$$

where $v_i$ denotes the occupancy probability of $i^{th}$ voxel in the grid $F_t^G$, $z_j$ is the measurement event that $j^{th}$ camera ray passes through this voxel. The numerator of the item $\log \frac{p(z_j|v_i=1)}{p(z_j|v_i=0)}$ means that the probability of being passed for a voxel if this voxel is occupied in fact, which shows the confidence of ray casting process. Obviously, the numerator and denominator can be set as an empirical constant. Therefore, we update the log-odds occupancy probability of each voxel in the grid $F_t^G$ by adding a constant one time when a single camera ray passes through this voxel. Note that the probabilistic occupancy grid $F^G$ is continually updated within one episode. Finally, the occupancy status of voxels can be classified into three categories: unknown, occupied, and free, by setting a probability threshold.

We implement this algorithm with PyTorch. In particular, we used ray-casting renderers from PyTorch3D [22] to generate and use depth maps as inputs to our model.