# OpenReview forum: "GenNBV: Generalizable Next-Best-View Policy for Active 3D Reconstruction"
_ICLR.cc/2024/Conference — ICLR 2024 Conference Withdrawn Submission_

### Official Review · Reviewer_k3Gz · 2023-10-25

**Soundness:** 1 poor
**Presentation:** 1 poor
**Contribution:** 1 poor
**Rating:** 3
**Confidence:** 5

**Summary:**

The paper proposes an end-to-end framework to train an RL policy that maximizes the coverage ratio for exploration. Though targeting an interesting task, the authors fail to highlight their core contributions and the novelty is quite limited. Using probabilistic occupancy grids for autonomous reconstruction has actually been well-studied for quite a long time. The experiments are not sufficient to validate the efficacy of the proposed method. Meanwhile, the comparisons against NeRF-based methods instead of the conventional grid-based NBV methods are confusing.

**Strengths:**

The results in Tab. 2 are interesting. How different factors in the unimodal and multimodal settings affect the final results are worth studying with in-depth analysis.

**Weaknesses:**

1. Confusing setting.
- The title indicates that the proposed method is designed for '3D reconstruction'. However, the evaluation metrics only evaluate the completeness/coverage ratio but not the reconstruction accuracy. It is unclear how the reconstructed scenes are visualized in Fig. 1-4 and the supplementary video. The grid size of the map is also unclear.
- It is unclear why the paper mentions and compares against the NeRF-based methods as the proposed method utilizes conventional voxel grids as the scene representation. Using 3D voxel grid for next-best-view planning is well-studied, but none of the relevant papers appear in the [Related Work] section nor the [Experiments] section.

2. Unjustified property of "generalization".
- There are plenty of strong arguments indicating that existing methods fail to generalize (page 1&4) without supported experimental results. No evaluation regarding the generalization ability is conducted in the [Experiments] section.

3. Unclear novelty of the proposed method.
- Most contents regarding the formulation, representation, and rewards in the [Methodology] section are commonly used. It is unclear what is the key contribution/novelty of the proposed method.

**Questions:**

None.

---

### Official Review · Reviewer_NscJ · 2023-11-01

**Soundness:** 2 fair
**Presentation:** 2 fair
**Contribution:** 1 poor
**Rating:** 3
**Confidence:** 3

**Summary:**

GenNBV is presented as a new framework for the Next-Best-View (NBV) problem, emphasizing end-to-end training for active 3D reconstruction. The NBV task is redefined as a reinforcement learning task, with the introduction of a policy network that leverages a coverage ratio for its reward function. This approach aims to deduce near-optimal NBV for unfamiliar structures. Comparative studies with various datasets demonstrate its superior performance over existing methodologies.

**Strengths:**

- The paper formulated the next-best-view task in the context of reinforcement learning by defining state, action, and reward.
- An RL framework specifically tailored for the NBV problem is introduced taking images and actions for training policy net that predict next best view for 3D reconstruction.

**Weaknesses:**

- Limited Technical Novelty
  -The paper's technical novelty appears constrained, primarily focusing on presenting the NBV as a reinforcement learning task. Notably, this is not the first work to do so, with Scan-RL having previously introduced RL-based approach for NBV. Further, the proposed RL framework doesn't markedly deviate from Scan-RL's approach, with the primary distinction being the geometric representation derived from depth maps.
- Insufficient Experimental Evidence
  - The paper lacks a comprehensive set of experiments or in-depth analyses that highlight its advantages.
  - As per Table 1, Scan-RL's performance closely mirrors that of the proposed method. Given that Scan-RL doesn't incorporate depth information—a key geometric representation in this study—the slight performance differential might be attributed to the inclusion of depth modality.
- The paper relies heavily on empirical values (thresholds, $C$), raising concerns about their applicability across diverse datasets. A more thorough explanation or rationale for these values would be beneficial.

**Questions:**

- Can the empirical constant C and the threshold set for the occupancy map be universally applied across varied datasets?

---

### Official Review · Reviewer_7e87 · 2023-11-09

**Soundness:** 3 good
**Presentation:** 3 good
**Contribution:** 3 good
**Rating:** 6
**Confidence:** 2

**Summary:**

This paper focuses on generalizable NBV prediction for active 3D reconstruction via RL. Without limitations on the action spaces of agents, the proposed method uses a drone (RGB-D and IMU input) to scan and reconstruct the underlying outdoor scenes via probabilistic occupancy grids. Experiments on synthetic data demonstrates the effectiveness of GENNBV on unseen and novel outdoor scenes.

**Strengths:**

- The overall paper is well motivated and written, which is straightforward to follow.
- Experiments on Houses3K, OmniObject3D datasets shows that there is a performance gain compared to recent baselines of active reconstruction.

**Weaknesses:**

Though achieving promising generalization capability on novel synthetic scenes, I still have a few concerns towards the evaluation and practicability of GENNBV.
1. The experiments are limited in synthetic set-up, which is reasonable considering the RL pipeline. However, there is no practical demonstrations of how this would transfer to real-world reconstructions when the dynamic of agents, the captured RGB-D frames and poses will be imperfect and suffer from their physic limitations.
How does the methods work given imperfect input?

2. About evaluation, I still think it would be more valuable providing more evaluations given diverse number of views ranging from an extreme case of 1-3 to a adequate large number of views like 100. This would demonstrate the performance curve giving different viewing budgets and tell the readers when does the proposed method tends to saturate its performance.

3. An informative illustration would be the visualization of planned path compared to baselines. It will convince the readers whether the free-space action capability really matters and bring different trajectories compared to a classifical hemi-sphere one or so. Or the planned trajs are similar to some extent?

4. Does the proposed methods adapt to different underlying reconstruction method (explicit prob girds, TSDF-grids, NeRF-like volumetric implicit fields, etc)? Some recent work like Lee et al and NeurAR uses network predicted uncertainty. Will such network inferred probs merge with the odds of occupancy maps of GENNBV?

Therefore, I am slightly above borderline but would like to adjust my socre based on authors' feedbacks.

**Questions:**

Please see the weaknesses section above.

---

### Official Review · Reviewer_gNx2 · 2023-11-10

**Soundness:** 2 fair
**Presentation:** 2 fair
**Contribution:** 2 fair
**Rating:** 5
**Confidence:** 4

**Summary:**

## Summary
This paper proposes a new framework for active 3D reconstruction. A probabilistic 3D occupancy grid is used as the mapping backbone. The major contribution of this paper is a reinforcement learning-based next-best-view policy. The authors proposed a dataset that allows the agent to learn RL policy from diverse scenes. An embedding strategy that considers action, geometric, and semantics, is included to boost the result. This paper has shown better results than some baseline methods.

**Strengths:**

### Improved generalization ability for NBV learning
The authors have proposed an RL policy that trains from diverse 3D object simulations for NBV. With the trained policy, the authors have shown that it can be generalized to a new dataset with real-world collection.

### Ablation studies to support the embedding strategy
The authors have provided an ablation study to show the effectiveness of the proposed embedding strategies. The result shows that the proposed multimodal representation helps the final reconstruction quality.

### Non-predefined action space->generalization ability (free motion)
The proposed method also allows a free 3D action space which differs from most of the prior work. It allows the usage of more general scenes.

**Weaknesses:**

### Missing related work
NBV is not a task with learning-based methods. This paper mainly discusses recent works that use the radiance field as the mapping module. However, there are a bunch of works that use classical methods for NBV, e.g. [A][B]. I would suggest the authors refer to [C] for a more detailed survey of this field.
[A] A comparison of volumetric information gain metrics for active 3d object reconstruction
[B] An information gain formulation for active volumetric 3d reconstruction
[C] A Survey on Active Simultaneous Localization and Mapping: State of the Art and New Frontiers

### Misleading statement: first free space NBV policy
There are prior works for free space NBV. The proposed method is not the first one. Please check [C] for the classical methods. For recent radiance field based methods, ActiveRMap and [D]  also consider free space motion.
[D] NeurAR: Neural Uncertainty for Autonomous 3D Reconstruction with Implicit Neural Representations

### Unfair comparison
The prior works listed in the comparison table are RGB-based methods. This proposed system relies on RGB-D inputs. With depth sensor, it is supposed to have better reconstruction quality.
It would be a more fair comparison against methods with depth sensing.
[A][B] should be a good reference. Authors can find more methods in [C] for a valid comparison. For recent radiance/uncertainty based methods, [D] also uses depth maps.

**Questions:**

### Generalizability question
In this paper, the authors have included a larger motion space. However, the simulations are outside-in scenes (correct me if I am wrong). The learned policy is learned specifically for this kind of scanning.
Does the policy generalize to other scenes, e.g. inside-out scenes?
Even if a real-world dataset is applied, the scanning is under same outside-in setup.